# AN IMPORTANT THING TO DO BEFORE FEDERATED TRAINING

**Yichu Xu**[1*]**, Wenqian Li**[2*]**, Yinchuan Li**[3†]**, Yunfeng Shao**[3]**, Yan Pang**[2]**, De-Chuan Zhan**[1]

[1]State Key Laboratory for Novel Software Technology, Nanjing University
[2]National University of Singapore, Singapore
[3]Huawei Noah's Ark Lab, Beijing, China
{xuyc,zhandc}@lamda.nju.edu.cn, Wenqian@u.nus.edu,
{liyinchuan,shaoyunfeng}@huawei.com, jamespang@nus.edu.sg

## ABSTRACT

Previous research in Federated learning (FL) have emphasized privacy protection, model optimization, and so on, meanwhile, they overlooked how to choose the appropriate FL algorithm for a new federation with preserving data privacy. In our study, we provide a formal problem formulation for algorithm selection in FL and present a novel approach that involves leveraging trained federations to aid with algorithm selection. Empirical results prove the effectiveness of our method.

## 1 INTRODUCTION

Federated Learning (FL) was proposed to learn a global model through multiple data owners (federation) who are coordinated by a centralized server without sharing raw data (McMahan et al., 2017). Researchers have emphasized communication efficiency, data heterogeneity, and so on. However, they overlooked an essential part of the whole FL service: what kind of FL algorithm should be used for a new federation without privacy leakage. Some studies have conducted performance evaluations of different FL algorithms and given some guides about how to select an appropriate FL algorithm according to the kind of data heterogeneity and the demand of federation (Nilsson et al., 2018; Mulay et al., 2020; Li et al., 2022). However, existing guides for selecting FL algorithms do not apply to unknown heterogeneous situations due to data privacy. Therefore, this paper proposes an under-explored problem in FL: how to choose the most suitable FL algorithm for an unseen particular federation? Then, we design a novel method for selecting the optimal FL algorithm for a federation. To the best of our knowledge, this is the first work to propose this direction in federated learning, and the pipeline is shown in Figure 1.

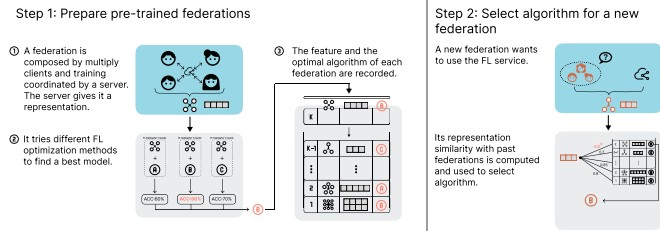

Figure 1: Pipeline for Algorithm Selection

## 2 METHODOLOGY

Suppose there is a set of trained federations $\mathcal{K}$, where $|\mathcal{K}| = K$. Each federation $k \in \mathcal{K}$ has a client set $\mathcal{N}_k$ and the data set $\mathcal{D}_k$, where $|\mathcal{N}_k| = n_k$ represents the number of clients in that

---

*Equal Contribution
†Corresponding Author: Yinchuan Li.

federation. Within each federation $k$, we define the client $\forall i \in \mathcal{N}_k$ has the data set $\mathcal{D}_{k,i}$ and thus $\mathcal{D}_k = \{\mathcal{D}_{k,i}\}_{i=1}^{n_k}$. With an assumption that $\forall k \in \mathcal{K}$, we have its best FL algorithm $\mathcal{A}_k^*$, which trains the model achieving the highest accuracy on the global test set $\mathcal{D}_{\text{test}}$ among various algorithms. Given a new federation $j \notin \mathcal{K}$ without a trained model, our objective is to find the best FL algorithm $\mathcal{A}_j^*$ based on the federation set $\mathcal{K}$.

It is straightforward for federation $j$ to try all of the algorithms in $\{\mathcal{A}_k^*\}_{k=1}^K$ and choose the one achieving the best accuracy on the test data set. However, this brute-force comparison makes the process expensive in computation and communication. In intuition, the performance of the model averaged from two clients could reflect the relationship between them. For example, the aggregated performance of two clients with similar data distributions is better than two clients with heavily heterogeneous data distributions. It shows the aggregated performance of clients is related to the relationship between data distribution. Therefore, it is feasible to leverage statistical information of the aggregated performance of various client subsets to express the inner characteristic of a federation, concretely, the discrepancy of clients, which can be considered as **federation representation**. Through federation representation, our goal is to find the most similar federation in $\mathcal{K}$ and its algorithm for the new federation $j$.

Let $V_k^t(S)$ denote the performance of aggregated model from a client subset $S \subseteq \mathcal{N}_k$ in the $t$-th round. We define $U_k^{(\ell)}$ as the variance of performance of all subsets with $\ell$ clients in federation $k$ such that $U_k^{(\ell)} = \text{Var}\left(\{V_k^t(S), S \subseteq S^{(\ell)}\}\right), 1 \leq \ell \leq n_k$, where $S^{(\ell)} = \{S, S \subseteq \mathcal{N}_k \wedge |S| = \ell\}$, and the statistics are incorporated as representation of federation $k$ as $U_k = [U_k^{(1)}, U_k^{(2)}, \cdots, U_k^{(n_k)}]$.

Given $K$ target federations $\{U_k, \mathcal{A}_k^*\}_{k=1}^K$, and a query federation $j$ with $U_j$, we measure the similarity between federation $j$ and $k$ via $d(U_j, U_k) = \sum_{\ell=1}^m U_j^{(\ell)} \times U_k^{(\ell)} / \sqrt{(\sum_{\ell=1}^m (U_j^{(\ell)})^2) \times (\sum_{\ell=1}^m (U_k^{(\ell)})^2)}$, where $m = \min(n_j, n_k)$. Then, we choose the algorithm for federation $j$ based on $\mathcal{A}_j^* = \arg\max_{\mathcal{A}_k^*} d(U_j, U_k)$. However, the time complexity of computing $U_k$ could be $O(2^{n_k})$ which is a very expensive cost as $n_k$ increases. Therefore, we introduce a threshold $\tau$ to limit the maximal size of the set $S^{(\ell)}$. Specifically, if the size of set $S^{(\ell)}$ is larger than $\tau$, we sample $\tau$ subsets from $S^{(\ell)}$ to decrease computation costs.

## 3 EXPERIMENT RESULTS

To simulate different federations, we combined two datasets (CIFAR-10 and CIFAR-100s) and two data splits (dirichlet sampling and label sampling) resulting in four federations F1-F4. The federation F1 and F2 which is constructed on CIFAR-10 are considered as target federations whose best algorithm has been identified by exhaustive searching among three

|  | Query F3 ▲ | F4 ▼ |
|---|---|---|
| Target | | |
| F1 ▲ | **0.956**(FedDyn) | 0.890 |
| F2 ▼ | 0.916 | **0.973**(FedRS) |

Table 1: The results of algorithm selection.

algorithms including FedAvg (McMahan et al., 2017) , FedDyn (Acar et al., 2021) and FedRS (Li & Zhan, 2021), while F3 and F4 are constructed on CIFAR-100s and considered as query federations. ▲ and ▼ denotes dirichlet sampling and label sampling, respectively. Following (Acar et al., 2021; Li & Zhan, 2021), we uniformly use a two-layer convolutional network (LeCun et al., 1998). More details are shown in Appendix A.2. As shown in Table 1, our method can correctly match the query federation with the target federation whose data distribution is as same as its. Through trying out three algorithms, the best algorithm of the query federation is as same as the one of matched target federation. It proves that our method can help a federation to choose an appropriate algorithm.

## 4 CONCLUSION

We propose an under-explored problem of selecting a suitable algorithm for a new federation without privacy leakage. To accomplish this, we extract query and trained federations representations by utilizing the statistics of aggregated performance from various client subsets within the federation. Then, we select the algorithm of the most similar federation to the query one. Our experiment results prove the feasibility of our method.

URM STATEMENT

The authors acknowledge that the first three authors of this work meets the URM criteria of ICLR 2023 Tiny Papers Track. The first author is 24 years old and non-white student. The second author is 24 years old and non-white student. The third author is 28 years old and non-white.

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

## A  APPENDIX

### A.1  RELATED WORKS

#### A.1.1  FEDERATED LEARNING ALGORITHMS

As society increasingly emphasizes privacy and safety of client data, it becomes difficult to learn a well-performed model due to insufficient data. To address this issue, federated learning was first proposed by (McMahan et al., 2017) to enable multiple clients to jointly train a shared model coordinated by a centralized server without exposing their local data. Following (McMahan et al., 2017), many researchers have made progress on improving the efficiency and effectiveness of algorithm (Li et al., 2020; Karimireddy et al., 2020; Acar et al., 2021; Li & Zhan, 2021) and preserving the privacy of client data (Agarwal et al., 2018; Zhu et al., 2020). (Li et al., 2020) claimed that FedAvg can not converge well when the data distributions of clients are heterogeneous, and proposed FedProx which adds a proximal loss term to FedAvg for alleviating this data heterogeneity. (Karimireddy et al., 2020; Acar et al., 2021) respectively introduced control variates and a dynamic regularizer to solve the misalignment of optimization goals between the global model and local models of clients. (Li & Zhan, 2021) proposed to "Restricted Softmax" to limit classification weights of missing classes in scenarios where there are missing classes in clients. Unlike the aforementioned works, we focus mainly on the initial stage of the federated process, which involves selecting an appropriate FL algorithm for a new federation.

### A.2  EXPERIMENTS

#### A.2.1  EXPERIMENT SETUP

To construct different federations, we adapt two datasets: CIFAR-10 and CIFAR-100s (Krizhevsky et al., 2009). We refer to previous works on FL (Acar et al., 2021) and use a two-layer CNN for CIFAR-10 and CIFAR-100s, both of which are used to image classification tasks. CIFAR-100s is created by randomly selecting ten classes from the original CIFAR-100 dataset. Moreover, we choose two non-IID data sampling ways including dirichlet sampling (Hsu et al., 2019) and label sampling (McMahan et al., 2017). The former samples the distribution of clients from a dirichlet distribution to ensure different label ratios on clients. The latter randomly assigns two classes to each client. The detailed information of federations is listed in Table 2 and the data distributions in each federation are shown in Figure 4.

In the part of extracting representation, we conduct 10 rounds via FedAvg and use $t = 10$ round to compute $V_k^t(S)$ for the representation of federation $k$. When round $t$ becomes larger, the statistics $U_k$ becomes more stable. For two target federations, we try out three FL algorithms including FedAvg (McMahan et al., 2017) , FedDyn (Acar et al., 2021) and FedRS (Li & Zhan, 2021). The best performance of FedAvg, FedDyn and FedRS on target federations are shown in Figure 2. The best performance of FedDyn and FedRS on query federations are shown in Figure 3

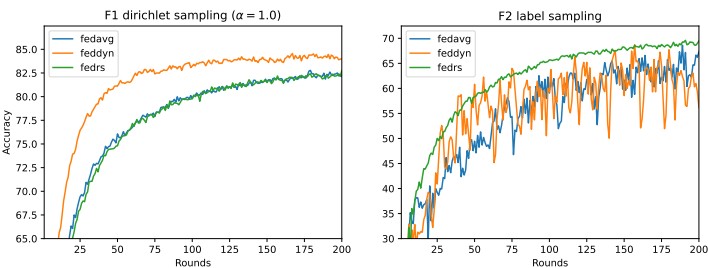

Figure 2: Performance on two target federations

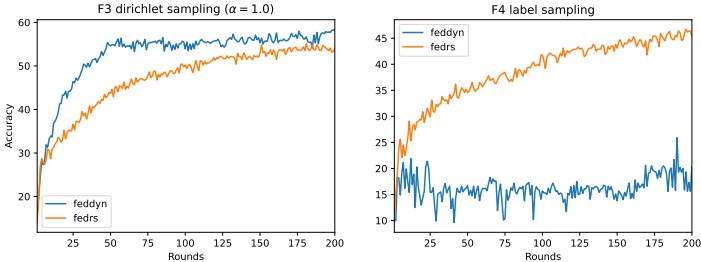

Figure 3: Performance on two query federations

| federation | client numbers | dataset | data split |
|------------|----------------|-----------|--------------------|
| F1 | 10 | CIFAR-10 | dirichlet sampling |
| F2 | 10 | CIFAR-10 | label sampling |
| F3 | 20 | CIFAR-100s | dirichlet sampling |
| F4 | 20 | CIFAR-100s | label sampling |

Table 2: Details of four federations

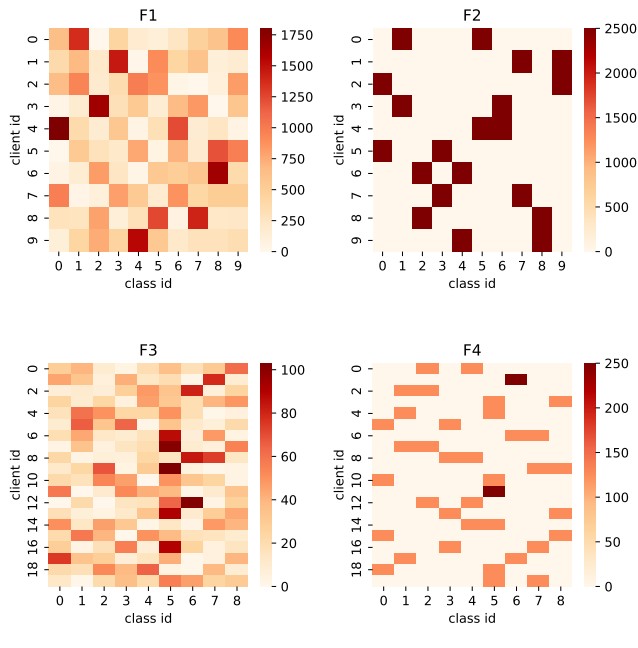

Figure 4: Detailed data distribution of four federations

