# OpenReview forum: "One Important Thing To Do Before Federated Training"
_ICLR.cc/2023/TinyPapers — Submitted to Tiny Papers @ ICLR 2023_

### Official Review · Reviewer_DndC · 2023-03-24

**Confidence:** 4

**Summary Of Contributions:**

The paper asks two questions to be resolved for an effective federated learning. In turn, the proposed research aism to provide an algorithm selection framework which can help in learning better algorithm in FL setting.

**Rating:**

Needs Clarification (NC): a submission which does not meet the reviewing criteria and needs clarification for its described problem or solution

**Strengths And Weaknesses:**

The questions asked although well known in the community and have received due attention recently are important to explore.

However, the paper needs significant writing updates. The abstract should be concise and clear. It did not give any idea about the questions and what has been proposed in this paper.

The contribution of the proposed work is very limited and has not been justified quantitatively. The weakness of the existing works which is overcome is also not well highlighted.

**Suggested Changes:**

Proofread the paper and update the paper to make it clear and concise. The paper is missing a point description which makes it hard to understand the limitation of existing works and the proposed contribution.

No analysis has been performed in this paper to justify the impact of this work.

Solutions to the questions raised are not clearly demonstrated. Even the questions themselves are not clear.

---

> ### Author Response · Authors · 2023-06-01
> **Response to Reviewer DndC**
>
> Dear Reviewer,
>
> Thank you for your constructive suggestions. They are really helpful for us to improve our paper.
>
> 1. **the questions are not clear**: We have opted to focus on a detailed definition of the problem of algorithm selection in FL instead of introducing two problems, due to the 2-page limit of the Tiny Papers.
> 2.  **hard to understand the limitation of existing works and the proposed contribution**: We have added a discussion of prior works which conducted performance evaluation of FL algorithms and provided guidelines for selecting FL algorithms based on known data distributions. However, these research often necessitate that clients divulge private information about their data distribution. In contrast, we propose the problem of selecting a suitable algorithm for a federation without privacy leakage. Our novel approach involves leveraging trained federations to assist with algorithm selection, and our experimental results demonstrate the feasibility of our proposed method.
> 3.  **No analysis has been performed in this paper to justify the impact of this work. Solutions to the questions raised are not clearly demonstrated.**: We have made revisions to our experimental results, providing a clearer and more concise explanation of our findings.
>
> Thanks for your suggestions! We hope that we have solved the mentioned weaknesses.
>
> References:
> - Qinbin Li, Yiqun Diao, Quan Chen, and Bingsheng He. Federated learning on non-iid data silos: An experimental study. In 38th IEEE International Conference on Data Engineering, ICDE 2022, pp. 965–978. IEEE, 2022
> - Ajinkya Mulay, Gaspard Baye, Rakshit Naidu, Santiago Gonz ́alez-Toral, Vineeth S, Tushar Semwal, and Ayush Manish Agrawal. Fedperf: A practitioners’ guide to performance of federated learning algorithms. In NeurIPS 2020 Workshop on Pre-registration in Machine Learning, volume 148, pp. 302–324. PMLR, 2020.
> - Adrian Nilsson, Simon Smith, Gregor Ulm, Emil Gustavsson, and Mats Jirstrand. A performance evaluation of federated learning algorithms. In Proceedings of the Second Workshop on Distributed Infrastructures for Deep Learning, DIDL@Middleware 2018, pp. 1–8. ACM, 2018

---

### Author Response · Authors · 2023-06-01
**Revised paper and request for archival**

Dear ICLR Program Chairs,

We have completed the latest revision of our paper, and we have responded to each of the reviewers’ comments.
Furthermore, we would like to request that our paper can be archived.

Thank you for your time and attention!

---

### Meta-Review · Area_Chair_8ue4 · 2023-04-07

**Recommendation:** Invite to revise
**Confidence:** 4

**Metareview:**

The paper asks pertinent questions that have been on the radar of the FL community in the recent times i.e. Algorithm Selection and Client Contribution Prediction for effective FL. This, IF executed well not only helps bring attention of the community to these sub-problems, but also puts a step forward in formalizing such efforts for further contribution.

However, the current state of the submission needs significant revisions to meet archivable standards.
These revisions include better formalization, experimentation, analysis and comparison to existing works to make a case for the paper and it's stand.

Overall, we recommend an invitation to revise for the submission.

**Summary:**

The paper highlights two supposedly under-explored directions in FL community, i.e. Algorithm Selection and Client Contribution prediction. It also proposes a framework for algorithm selection. The paper lacks in formalization, literature review and analysis.

**Comments And Feedback To The Authors:**

The paper asks pertinent questions that have been on the radar of the FL community in the recent times. This, if executed well not only helps bring attention of the community to these sub-problems, but also puts a step forward in formalizing such efforts for further contribution.

The current submission needs some revisions to reach above-mentioned standard:
1. The research questions proposed could be clearer and formalized.
2. It must disambiguate and attribute contributions from other works in the domain exploring similar topics.
3. Having some quantitative experiments to substantiate improvements through the suggested algorithm might be helpful in enforcing confidence.


**Reason For Not Giving A Higher Recommendation:**

Given, the available review and personal opinion, I conclude that the current state of the submission needs significant revisions to meet archivable standards. These revisions include better formalization, experimentation, analysis and comparison to existing works to make a case for the paper and it's stand.

**Reason For Not Giving A Lower Recommendation:**

N/A

---

> ### Author Response · Authors · 2023-06-01
> **Response to Area Chair 8ue4**
>
> Dear ICLR Area Chair,
>
> We appreciate your valuable feedback and meta-review. We provide our responses in the following section, which includes a restatement of your comments followed by our point-by-point responses.
>
> 1. **The research questions proposed could be clearer and formalized:** Upon careful consideration, we have decided to limit our focus on the research research question of algorithm selection due to the 2-page limit. Therefore, our modification and responses mainly pertain to this aspect.
> 2. **It must disambiguate and attribute contributions from other works in the domain exploring similar topics**: Based on our extensive review of the literature, prior research has conducted numerous experiments with different data distributions, evaluation metrics, and FL algorithms, leading to the identification of statistical relationships between them. However, these guidelines often assume that the server is aware of the data distribution of each client, increasing the risk of privacy violations. Additionally, previous works have not utilized federations that have already been trained, despite the possibility of a FL service provider assisting multiple federations in their model training efforts. Thus, taking into account these limitations, we have formally defined the problem of (FL) algorithm selection and proposed a practical solution to address these issues.
> 3. **Having some quantitative experiments to substantiate improvements through the suggested algorithm might be helpful in enforcing confidence:** We have modified our tables and figures about experiments in order to demonstrate the feasibility of our proposed method.
>
> Thanks again for your feedback! We hope that our revised paper have solved the above weaknesses.
>
> References:
> - Qinbin Li, Yiqun Diao, Quan Chen, and Bingsheng He. Federated learning on non-iid data silos: An experimental study. In 38th IEEE International Conference on Data Engineering, ICDE 2022, pp. 965–978. IEEE, 2022
> - Ajinkya Mulay, Gaspard Baye, Rakshit Naidu, Santiago Gonz ́alez-Toral, Vineeth S, Tushar Semwal, and Ayush Manish Agrawal. Fedperf: A practitioners’ guide to performance of federated learning algorithms. In NeurIPS 2020 Workshop on Pre-registration in Machine Learning, volume 148, pp. 302–324. PMLR, 2020.
> - Adrian Nilsson, Simon Smith, Gregor Ulm, Emil Gustavsson, and Mats Jirstrand. A performance evaluation of federated learning algorithms. In Proceedings of the Second Workshop on Distributed Infrastructures for Deep Learning, DIDL@Middleware 2018, pp. 1–8. ACM, 2018

---

> ### Author Response · Authors · 2023-06-01
> **Response to Area Chair 8ue4**
>
> Dear ICLR Area Chair,
>
> We appreciate your valuable feedback and meta-review. We provide our responses in the following section, which includes a restatement of your comments followed by our point-by-point responses.
>
> 1. **The research questions proposed could be clearer and formalized:** Upon careful consideration, we have decided to limit our focus on the research research question of algorithm selection due to the 2-page limit. Therefore, our modification and responses mainly pertain to this aspect.
> 2. **It must disambiguate and attribute contributions from other works in the domain exploring similar topics**: Based on our extensive review of the literature, prior research has conducted numerous experiments with different data distributions, evaluation metrics, and FL algorithms, leading to the identification of statistical relationships between them. However, these guidelines often assume that the server is aware of the data distribution of each client, increasing the risk of privacy violations. Additionally, previous works have not utilized federations that have already been trained, despite the possibility of a FL service provider assisting multiple federations in their model training efforts. Thus, taking into account these limitations, we have formally defined the problem of (FL) algorithm selection and proposed a practical solution to address these issues.
> 3. **Having some quantitative experiments to substantiate improvements through the suggested algorithm might be helpful in enforcing confidence:** We have modified our tables and figures about experiments in order to demonstrate the feasibility of our proposed method.
>
> Thanks again for your feedback! We hope that our revised paper have solved the above weaknesses.
>
> References:
> - Qinbin Li, Yiqun Diao, Quan Chen, and Bingsheng He. Federated learning on non-iid data silos: An experimental study. In 38th IEEE International Conference on Data Engineering, ICDE 2022, pp. 965–978. IEEE, 2022
> - Ajinkya Mulay, Gaspard Baye, Rakshit Naidu, Santiago Gonz ́alez-Toral, Vineeth S, Tushar Semwal, and Ayush Manish Agrawal. Fedperf: A practitioners’ guide to performance of federated learning algorithms. In NeurIPS 2020 Workshop on Pre-registration in Machine Learning, volume 148, pp. 302–324. PMLR, 2020.
> - Adrian Nilsson, Simon Smith, Gregor Ulm, Emil Gustavsson, and Mats Jirstrand. A performance evaluation of federated learning algorithms. In Proceedings of the Second Workshop on Distributed Infrastructures for Deep Learning, DIDL@Middleware 2018, pp. 1–8. ACM, 2018

---

### Decision · Program_Chairs · 2023-04-09

Revision accepted; invite to archive